# Receptor-Independent Therapies for Forensic Detainees with Schizophrenia–Dementia Comorbidity

**DOI:** 10.3390/ijms242115797

**Published:** 2023-10-31

**Authors:** Adonis Sfera, Luminita Andronescu, William G. Britt, Kiera Himsl, Carolina Klein, Leah Rahman, Zisis Kozlakidis

**Affiliations:** 1Paton State Hospital, 3102 Highland Ave, Patton, CA 92369, USA; andronescu.luminita@dsh.ca.gov (L.A.); kiera.himsl@dsh.ca.gov (K.H.); 2School of Behavioral Health, Loma Linda University, 11139 Anderson St., Loma Linda, CA 92350, USA; 3Department of Psychiatry, University of California, Riverside 900 University Ave, Riverside, CA 92521, USA; 4Department of Psychiatry, School of Medicine, Loma Linda University, Loma Linda, CA 92350, USA; wbeitt@llu.edu; 5California Department of State Hospitals, Sacramento, CA 95814, USA; carolina.klein@dsh.ca.gov; 6Department of Neuroscience, University of Oregon, 1585 E 13th Ave, Eugene, OR 97403, USA; leah.rahman@dsh.ca.gov; 7International Agency for Research on Cancer, 69366 Lyon Cedex, France; kozlakidisz@iarc.who.int

**Keywords:** cell membrane lipids, schizophrenia, neurocognitive disorders, psychotropic medication, dopamine hypothesis

## Abstract

Forensic institutions throughout the world house patients with severe psychiatric illness and history of criminal violations. Improved medical care, hygiene, psychiatric treatment, and nutrition led to an unmatched longevity in this population, which previously lived, on average, 15 to 20 years shorter than the public at large. On the other hand, longevity has contributed to increased prevalence of age-related diseases, including neurodegenerative disorders, which complicate clinical management, increasing healthcare expenditures. Forensic institutions, originally intended for the treatment of younger individuals, are ill-equipped for the growing number of older offenders. Moreover, as antipsychotic drugs became available in 1950s and 1960s, we are observing the first generation of forensic detainees who have aged on dopamine-blocking agents. Although the consequences of long-term treatment with these agents are unclear, schizophrenia-associated gray matter loss may contribute to the development of early dementia. Taken together, increased lifespan and the subsequent cognitive deficit observed in long-term forensic institutions raise questions and dilemmas unencountered by the previous generations of clinicians. These include: does the presence of neurocognitive dysfunction justify antipsychotic dose reduction or discontinuation despite a lifelong history of schizophrenia and violent behavior? Should neurolipidomic interventions become the standard of care in elderly individuals with lifelong schizophrenia and dementia? Can patients with schizophrenia and dementia meet the Dusky standard to stand trial? Should neurocognitive disorders in the elderly with lifelong schizophrenia be treated differently than age-related neurodegeneration? In this article, we hypothesize that gray matter loss is the core symptom of schizophrenia which leads to dementia. We hypothesize further that strategies to delay or stop gray matter depletion would not only improve the schizophrenia sustained recovery, but also avert the development of major neurocognitive disorders in people living with schizophrenia. Based on this hypothesis, we suggest utilization of both receptor-dependent and independent therapeutics for chronic psychosis.

## 1. Introduction

The population worldwide is aging at a rapid pace and so are forensic detainees with severe psychiatric illnesses, including schizophrenia (SCZ) and schizophrenia-like disorders (SLDs). In forensic institutions, there are two subpopulations of elderly patients with neurocognitive disorders: career criminals with lifelong mental illness and elderly first offenders without previous psychiatric history. The latter category often includes patients with frontotemporal dementia behavioral variant (bvFTD), a condition with SCZ-like clinical manifestations. As the incidence of dementias, including bvFTD, has increased in forensic population, screening all offenders over the age of 50 could identify these conditions early, ensuring adequate adjudication and placement.

At the cellular level, bvFTD has been associated with the selective loss of von Economo neurons (VENs), large cells, located in the anterior cingulate cortex (ACC), insular cortex (IC), and frontopolar cortex (FPC). As VENs pertain to the human neuro-moral network which plays a key role in amiability and empathy, loss of these cells is believed to engender a type of acquired sociopathy [1,2,3]. Interestingly, dysfunctional ACC, IC, and FPC have been linked to SCZ and impaired disease awareness (anosognosia). This pathology increases the odds of criminal violations and, several recent studies have suggested that anosognosia may be a better predictor of criminal behavior than focal brain lesions or clinical diagnoses [4,5,6,7,8,9,10]. 

Improved medical care, sanitation, and nutrition have contributed to increased longevity in both the population at large and institutionalized individuals with criminal history and chronic SCZ and SLDs [11,12,13]. On the other hand, increased lifespan led to a unique pathology, late-life dementia in people living with SCZ, a complex entity difficult to manage in forensic institutions [14,15]. The complexity of this phenomenon is further compounded by the long-term treatment with dopamine (DA)-blocking drugs, agents associated with serious adverse effects in patients with dementia [16,17,18,19]. This begets a clinical dilemma, as treatment of chronic psychosis can harm patients with dementia, while nontreatment may contribute to the re-emergence of violent behaviors. For this reason, new therapeutic strategies are urgently needed to address this conundrum, strategies that may involve neurolipidomic approaches, restoration of gut barrier homeostasis, and other receptor-dependent and independent antipsychotic treatments [20,21]. 

The available antipsychotic drugs excel at eliminating the positive symptoms in acute psychosis and they will likely remain the golden standard for the foreseeable future. However, exposure to these agents and chronic DA depletion was demonstrated to disrupt synaptic plasticity, contributing to impaired new information learning [22,23,24]. When superimposed on late-life dementia, dysfunctional neuroplasticity leads to more pronounced cognitive deficits, consistent with a distinct etiopathogenetic entity [25,26]. Indeed, under physiological circumstances, healthy aging was associated with increased synthesis of striatal DA, probably to compensate for the age-related downregulation of DA receptors [27,28,29]. Loss of synaptic plasticity, a modifiable risk factor, may be reversed with neurolipidomic therapies and/or DA reuptake inhibitors (DRIs) [30]. 

Antipsychotic drugs exert receptor-dependent and receptor-independent pharmacological actions. The former have been well-defined, while the latter are rarely mentioned in relation with these agents. Receptor-independent properties of antipsychotic drugs include antimicrobial, antiproliferative, pro-autophagy, and anti-endocytic characteristics, as well as lysosomotropism and peroxidation of plasma membrane lipids [31,32,33,34,35]. Indeed, several first- and second-generation antipsychotic drugs can disrupt neuronal and non-neuronal cells by inducing cell membrane lipid peroxidation. This, in turn, bends out of shape the cell surface, altering the receptor position and signaling as well as the transcellular transport of molecules. At the level of intestinal epithelia, antipsychotic-drug-mediated peroxidation of plasma membrane lipids, increases gut permeability, facilitating microbial translocation into host tissues [34,36,37,38,39,40] (Figure 1). SCZ has been associated with excessive lipid peroxidation and neuronal death by ferroptosis [41].

Another receptor-independent property of antipsychotic drugs is interference with iron metabolism [42,43]. As DA functions as a “de facto” iron chelator, it averts lipid peroxidation by sequestrating iron. Conversely, antipsychotic drugs upregulate iron, predisposing to neuronal death by ferroptosis [44,45]. On the other hand, DA can oxidize spontaneously in the presence of oxygen, generating superoxide radicals. Recently, a new group of phenazine analogs with antioxidant properties have been developed as anticancer therapeutics [46]. These iron-stabilizing, antioxidant compounds are natural (produced by microbes, including the gut flora), or synthetic [47]. To the best of our knowledge, these compounds have not been assessed for antipsychotic properties. Novel studies show that SCZ-induced gray matter loss could be averted by electron donating agents, such as phenoselenazine, one of the phenazine analogs [48,49,50]. 

In this article, we hypothesize that gray matter loss is the core symptom of schizophrenia which leads to dementia. We hypothesize further that strategies to delay or stop gray matter depletion would not only improve the schizophrenia sustained recovery, but also avert the development of major neurocognitive disorders in people living with schizophrenia. Based on this hypothesis, we suggest utilization of both receptor-dependent and independent therapeutics for chronic psychosis. Receptor-independent antipsychotic interventions include lipid replacement therapy (LRT), recombinant human IL-22, and aryl hydrocarbon receptor (AhR) antagonists. We also take a closer look at the dysfunctional interoceptive awareness, associated with gray matter loss, and its role in decreased illness insight (anosognosia). We conclude by recommending screening older first offenders for dementias to ensure proper sentencing and placement.

## 2. The Neuroscience of Criminal Behavior

For the past two centuries, the relationship between neuroscience and criminal law has been an uneasy one. Although neuroscience has made significant inroads into the courtroom, due to the complexity of criminal behavior, a crime cannot, without speculation, be directly corelated to a specific central nervous system (CNS) region or pathology [51,52,53,54]. For example, the suicide of the renowned actor Robin Williams cannot be attributed to the Lewy body disease (LBD), with which he had been diagnosed, because most patients with LBD do not take their lives [55]. Another example, Charles Whitman, an individual who committed mass murder in 1966 and had a brain tumor involving his amygdala, this pathology could not be reliably linked to the crime as individuals with similar conditions may not engage in violent acts [56]. Interestingly, gray matter loss, a characteristic of SCZ and poor illness insight, has been associated with violent crime [57].

### Interoceptive Awareness and Anosognosia

Interoceptive awareness refers to the ability of perceiving the internal state of the body, including heart rate, respiration, position, and belonging of limbs. It is believed that human self-awareness or insight are driven by this system. 

In various pathologies, including strokes, interoceptive awareness is impaired, leading to anosognosia. For example, neurological deficits can give patients the false impression that limbs do not belong to them, a form of anosognosia, known as one-sided neglect [58]. SCZ, bvFTD, and chronic traumatic encephalopathy (CTE) have been associated with anosognosia as various brain areas may misprocess information, generating erroneous beliefs and delusions. This suggests that anosognosia may drive psychosis, rather than the other way around [59,60,61]. For example, being told by others that they are ill but not perceiving the illness themselves, may lead to formulating alternative explanations often involving unseen interference. 

In the early stages, bvFTD may mimic SCZ or bipolar disorder as patients often exhibit impulsivity, apathy, lack of empathy, personality changes, and antisocial behavior. In addition, as memory may be intact for a long time, patients may score normally on mini mental status exam (MMSE), but less well when the executive function is being assessed. For this reason, bvFTD is often missed and patients misdiagnosed, frequently leading to incarceration. 

Selective loss of von Economo neurons (VENs) has been associated with dysfunctional interoceptive awareness and bvFTD, suggesting that these neurons likely drive anosognosia. As poor illness insight is found in up to 98% patients with SCZ, a characterized by gray matter loss, probably including the VENs [62,63,64,65]. Moreover, as anosognosia is a cognitive symptom, SCZ may be primarily a cognitive disorder with psychosis as a secondary manifestation, an observation made by Emil Kraepelin more than a century ago [66].

Anosognosia became the focus of research during the COVID-19 pandemic, which, like HIV before it, was associated with poor insight into cognitive impairment as well as gray matter loss [67,68,69,70]. Are VENs targeted by these viruses? At present, this question cannot be reliably answered, however, there is circumstantial evidence that this may be the case. For example, von Economo encephalitis, a viral infection which affected Europe and North America starting in 1915, resulted in criminal sequelae, involving many surviving children which were institutionalized because of criminal behavior [71,72]. Moreover, sporadic cases of this infection, reported more recently, have been associated with behavioral disturbances, including criminal behavior [73,74].

## 3. Schizophrenia as a Segmental Progeria

Segmental progeria, as SCZ is often called by researchers, is a syndrome of accelerated cellular aging marked by shortened telomeres that drive both neurodegeneration and all-cause mortality [75,76]. As a result, people living with schizophrenia have an average lifespan that is 15–20 years shorter compared to their healthy counterparts. However, due to improved care over the past three decades, the lifespan of this population has increased considerably, predisposing to neurodegenerative pathology [77,78]. Cellular senescence is an anticancer program of proliferation arrest, resistance to apoptosis, and active metabolism which is marked by a proinflammatory secretome, named senescence-associated secretory phenotype (SASP), which disseminates senescence to the neighboring healthy cells [79,80]. Senescence may explain the low-grade inflammation that plays a major role in SCZ onset and maintenance. Indeed, SCZ-associated premature cellular senescence affects all tissues and organs, including the cardiovascular system, gut, and blood–brain barrier (BBB), likely accounting for increased microbial translocation from the gastrointestinal (GI) tract into host systemic circulation [81]. In this regard, lipopolysaccharide (LPS), a cell wall component of Gram-negative bacteria, was detected postmortem in Alzheimer’s disease (AD) brains, suggesting migration from the gut through the senescent intestinal barrier [82]. This is significant as impaired intestinal permeability was documented in SCZ, explaining the high comorbidity of this disorder with inflammatory bowel disease (IBD) [81,83].

### Microbial Translocation

Upon CNS entry, microbes, or their components, including LPS or microbial cell-free DNA (mcfDNA), activate microglia, converting these brain macrophages into neurotoxic cells known to engage in the elimination of healthy neurons, probably including VENs [84]. Furthermore, a recent study detected plasma antibodies against specific gut microbes, including *Hafnei alvei, Pseudomonas aeruginosa, Morganella morganii, Pseudomonas putida,* and *Klebsiella pneumoniae* in patients with negative SCZ symptoms, linking this condition to microbial translocation [85,86]. 

Treatment with antipsychotic drugs affects neuronal cells via receptor-dependent and receptor-independent mechanisms. The latter include, among other changes, peroxidation of cell membrane lipids and intercalation of the drug in neuronal membrane’s lipid bilayer [37,87,88,89] (Figure 1). Moreover, as both SCZ and antipsychotics can induce peroxidation of cell membrane lipids, senescence-upregulated iron may trigger neuronal loss by ferroptosis [20,90].

We surmise that, due to their large cell surface, VENs are vulnerable to lipid peroxidation, explaining the preferential loss of these cells in bvFTD and gray matter loss in SCZ [91]. Moreover, mcfDNA, derived from the disintegration of translocated bacteria, induces Tau hyperphosphorylation and its accumulation in neurons, including VENs, leading to the selective demise of these cells [3,92]. This may be significant as anti-ferroptotic agents, as well as iron chelators, may comprise viable treatment options for bvFTD and other dementias [93,94,95].

## 4. Schizophrenia Outcome Studies—Kraepelin Was Right!

Numerous studies have shown disappointing progress in SCZ treatment, accounting for the continued need for long-term psychiatric facilities, such as state hospitals. For example, a century ago, there were large public institutions for tuberculosis, leprosy, and mental illness, while today only the latter remain in existence [52]. This demonstrates that sustained recovery in patients with SCZ or SLDs remains inadequate. There is no doubt that antipsychotics, the cornerstone therapeutics for acute psychosis, have contributed to improved symptom resolution and partial recovery, However, what patients desire the most is functionality, the ability to work or go to school, raise a family, and be independent in activities of daily living. These goals are rarely achieved by patients with chronic SCZ or SLDs, indicating that the currently available treatments do not address the core symptoms of this disorder [52,96,97,98]. For example, 33% of SCZ patients relapse during the first 12 months after an initial psychotic episode, 26% remain homeless at 2 years follow-up, while 5 years after the first psychotic outbreak, only 10% are employed [99,100,101]. Overall, 13.5% of SCZ patients meet recovery criteria at any point in time after the first psychotic episode [101]. These outcome studies are in line with a large metanalysis by Warner, R who looked at the entire 20th century and found that prognosis today is not very different compared to the early 20th century [54,102] (Table 1).

Furthermore, contrary to the expectations of most clinicians, the recovery rate following the introduction of antipsychotic drugs was not very different compared to the pre-psychopharmacological era [53]. Moreover, other studies have shown that employment of SCZ patients has decreased steadily over the past decades, a finding in line with neuroimaging studies which show lifelong gray matter loss, causing progressive disability [72,103] (Figure 2). 

Together this data suggests that “there may be an elephant in the room” that few researchers and clinicians talk about—gray matter loss. Indeed, progressive cortical tinning in SCZ, which seems to occur regardless of antipsychotic treatment, may account for the continuous existence of long-term institution for chronic mental illness, the disappointing recovery, and the progression of SCZ toward cognitive deficits, and disability [105,106,107] (Figure 2). Moreover, anosognosia, the cardinal SCZ symptom, rarely influenced by the antipsychotic drugs, was associated to gray matter loss in several brain regions, including the ACC, suggesting that this pathology may be the root cause of SCZ. [108]. We believe that slowing or stopping gray matter loss should be the goal of future SCZ treatments.

## 5. The Molecular Basis of SCZ and Dementia: Tau Protein Loss of Function

SCZ and dementia meet at the molecular level where they are marked by the instability of microtubules (MT). Recent studies demonstrate that SCZ is characterized by decreased microtubule-associated tau protein, which functions as a “molecular velcro”, holding the MTs together [109] (Figure 3). 

MTs are components of cellular cytoskeleton that, aside from maintaining the cell shape, participate in awareness and insight, suggesting that at the molecular level, anosognosia in SCZ may be the result of unstable MT due to decreased Tau [110,111,112]. Indeed, anosognosia in SCZ may be caused by dysfunctional MTs in VENs, cells with long axons that rely on MTs for intracellular transport of nutrients. Along this line, the role of MTs in cognition, self-awareness, and insight is further substantiated by anesthesia as anesthetics bind MTs at the tau site, likely destabilizing these molecules. Removal of tau may account for both loss of awareness during anesthesia and anosognosia [113,114]. 

### Chemo-Brain and Transplants

Another example of tau-mediated insight impairment, is “chemo-brain”, a phenomenon resulting from treatment of cancer patients with MT-targeting drugs, such as taxol, which is often followed by changes in cognitive and thought process that may lower insight, linking further MTs to cognition and interoceptive awareness [115]. 

Tau and MTs are expressed in other tissues, including the heart, which is known for possessing its own cognizance, distinct from the cortical awareness and memory [116,117]. For example, it has been established that following heart transplants, recipients often acquire not only select memories but also personality traits of the donor, further linking MTs to awareness, insight, and recall [118,119,120,121]. Indeed, tissue and cellular memory have been described in fascia, muscle, and gut microbiota, linking higher intellectual functions to MTs and tau [122,123]. Furthermore, gut microbes contain primitive MTs and can transfer learned behavior from one cell to another, indicating that rudimentary recall may be mediated by the MT and exported along with these molecules to other cells [124]. Along this line, translocated microbes may ‘instruct” host cells in tau aggregation as microbial DNA drives pathological tau [125]. Viruses, including SARS-CoV-2, can spread tau aggregates or neurofibrillary tangles (NFTs) from cell-to-cell in a prion-like manner, indicating that microorganisms may alter cognition, insight, and interoception [126] (Figure 3). Furthermore, upregulated intracellular iron in senescent cells can also promote tau aggregation, predisposing to neurodegeneration as well as to the loss of VENs by ferroptosis.

As opposed to dementia, including bvFTD, SCZ lacks NFTs, however, loss of tau protein can disrupt information processing due to MT destabilization [112,125,127,128]. Another VENs vulnerability: the lysosomes in these neurons exhibit high affinity for NFTs, possibly leading to cell death by excessive accumulation of proteine aggregates [129]. 

Taken together, the molecular underpinnings of cellular cognition appear to be present not only at cellular and organ level but also in subcellular and molecular structures. For example, “quantum mobility” in MT was suggested to drive awareness and cognition [130].

## 6. bvFTD: From Insight to Psychopathy

bvFTD is a common, early-onset dementia and, like other neurocognitive disorders, is a clinical syndrome that can be easily confused with primary psychiatric disorders, an error that in a forensic setting can complicate patients’ legal status, placement, and sentencing [131,132]. The clinical characteristics of bvFTD include early onset (compared to AD), criminal behavior (reported in 37–57% of patients), and neuropsychiatric symptoms, including apathy, disinhibition, and compulsions [133,134]. An interesting, and at the same time confusing feature of bvFTD consists of criminal violations without overt memory loss, at least during the early disease phases, a puzzling situation for forensic evaluators who may not suspect a neurodegenerative disorder [135]. 

Lack of specific neuroimaging and laboratory biomarkers, especially in the early disease stages, adds to the challenges of differentiating this syndrome from SCZ, bipolar disorder, or major depressive disorder (MDD) [136]. Indeed, over half of bvFTD patients had psychiatric diagnoses prior to the emergence of clear neurodegenerative signs, an error interfering with the adequate management and court proceedings [137]. 

The prevalence of bvFTD in forensic settings has increased over the past decade (measured by the surge of late-life first offenders), highlighting the importance of educating clinicians in keeping a high level of suspicion in first offenders over the age of 50. It is also crucial for state hospitals to capture data on the number of older first offenders. 

The correct diagnosis in forensic settings has a significant bearing on the adjudication and sentencing, as the judicial approach to criminal violations in individuals with neurodegenerative disorders is usually different than the classical “insanity defense” [138]. 

We recommend that all first offenders 50 years of age or older be screened for bvFTD. Screening should be accomplished by a neuropsychological profile, which usually reveals executive impairments and relative sparing of memory and visuospatial functions. Microbial translocation component can be evaluated by mcfDNA Karius Test^®®^ and IL-22 Singulex-Erenna^®®^ (Table 2). 

In addition, a task force should be established at local or state level, comprised of neuropsychologists, psychiatrists, and medicine/neurology professionals to screen for bvFTD and report to courts for appropriate action. Along this line, in 2015 at Patton State Hospital, California, we initiated a program for educating our clinicians on bvFTD via presentations, case reports, grand rounds, and publications to help identify this pathology early, if possible, at admission [140].

(https://www.reuters.com/article/us-crime-dementia/breaking-the-law-may-be-a-sign-of-dementia-idUSKBN0KE1Q020150105 (accessed on 29 October 2023). This idea was later adopted by other groups and clinician education in bvFTD continues to be implemented in various institutions and jurisdictions [138].

Like bvFTD, SCZ is a syndrome that in the second half of life is characterized by a “stable phase” with negative and cognitive symptoms, resembling dementia (Figure 2). This complicates not only the differential diagnosis but also the psychopharmacological management of SCZ with overlapping dementia. Indeed, comorbidity of SCZ with bvFTD or other dementias poses special challenges to the clinician, as antipsychotic drugs are associated with serious, even fatal, adverse effects when administered to dementia patients [141,142]. In this regard, clinicians may find themselves in a catch-22, as continuing antipsychotic treatment in SCZ patients and comorbid dementia may risk untoward effects, while discontinuing this therapy may lead to the reemergence of the dangerous behaviors. This dilemma calls for an interdisciplinary approach and involvement of hospital bioethics committee.

## 7. Neuropathological Basis of bvFTD

VENs, numbering about 193,000 cells, are large, corkscrew neurons located in layer V of the IC, ACC, and FPC. These large, non-telencephalic cells project to various brain areas and participate in prosocial cognition, empathy, and emotional intelligence. As parts of the salience network (SN), VENs respond to endogenous or exogenous stimuli in the order of priority [69,132]. 

SN dysfunction, documented in psychopathy and criminal behavior, connects this neuronal assembly to delinquency; however, depending on the jurisdiction, this evidence may not be allowed in a court of law [133]. Interestingly, a recent study connected the SN with gut microbes *Prevotella* and *Bifidobacterium*, emphasizing that the microbiome can influence the function of VENs, as well as their behavior, probably via MTs [143,144,145]. In addition, both *Prevotella* and VENs have been associated with suicidal behavior, further linking violence to dysfunctional gut permeability and microbial translocation outside the GI tract [146]. 

The homeostasis of intestinal barrier is maintained by interleukin 22 (IL-22), a cytokine regulated by aryl hydrocarbon receptor (AhR), a gut barrier protein responding to exogenous and endogenous ligands [63]. In this regard, VENs-rich IC was demonstrated to keep the record of gut inflammations, emphasizing the tight link between the GI tract and insula [147,148]. 

## 8. Dementia in People Living with Schizophrenia (PLWS), Potential Biomarkers 

A novel marker of microbial translocation, mcfDNA, is likely to become a diagnostic tool for evaluating dementia in PLWS, while at the same time providing proof that microbes can drive neuropathology [125,149,150]. For example, the commercially available assay mcfDNA Karius Test^®®^ detects peripheral blood microbial DNA, a molecule demonstrated to drive tau aggregation, predisposing a person to dementia [92,125] (Table 2). Conversely, SCZ is characterized by downregulated tau, indicating that this laboratory value could differentiate between SCZ with or without dementia [151,152,153,154]. For example, blood levels of phosphorylated tau (pTau) 217 and 181, elevated in dementia but normal in SCZ, can be helpful for differential diagnosis.

Another assay for gut barrier integrity available on the market, IL-22 Singulex-Erenna^®®^, measures IL-22 levels, a cytokine inversely corelated with gut barrier permeability and microbial translocation (Table 2). For example, low IL-22 with normal mcfDNA would reflect increased risk of microbial translocation without overt pathology. Conversely, low IL-22 and elevated mcfDNA would characterize frank bacterial translocation, requiring treatment. Interestingly, most antipsychotic drugs exert antibiotic properties, suggesting that blocking postsynaptic DA signaling may not be the only action mechanism of these agents. Moreover, IL-22 and mcfDNA could be useful for predicting the risk of relapse following discontinuation of antipsychotic drugs in forensic detainees.

Another marker that could differentiate SCZ from dementia is ferritin (elevated in dementia and low in SCZ), implicating iron dysmetabolism in both syndromes [20,155]. Excessive intracellular iron associated with senescent cells, a predominant SCZ phenotype, was shown to induce tau aggregation, implicating this biometal in tauopathies, including bvFTD [156,157]. Indeed, quantitative assessment of iron deposition is a new neuroimaging biomarker of bvFTD that can differentiate this phenotype from other FTD subtypes [158,159]. This suggests that in bvFTD, VENs may be lost through ferroptosis, a nonapoptotic cell death caused by iron-induced lipid peroxidation in the absence of glutathione-associated antioxidants. This is significant as intranasal deferoxamine was found effective for several neurodegenerative disorders, including bvFTD [160]. 

## 9. Chronic Traumatic Encephalopathy

A condition bvFTD-like condition, chronic traumatic encephalopathy (CTE), is a pathology associated with repeated traumatic brain injuries (TBIs), manifested by behavioral changes, including irritability, apathy, memory loss, and intermittent disorientation [161,162,163]. Upregulated iron and pTau may be the common denominators of bvFTD and CTE, emphasizing that neurodegeneration can be initiated by a localized or more generalized and diffuse pathology [164,165]. Along this line, the former NFL player, Phillip Adams, who killed six people, and committed suicide while serving his sentence, was found at autopsy to have had CTE, a condition some clinicians believed should have been factored in sentencing.

## 10. Interventions: Receptor-Independent Antipsychotic Treatments

The currently available antipsychotic drugs are extremely effective for the treatment of acute psychotic disorders, however, their efficacy in chronic psychosis is much less dramatic and may even predispose to neurocognitive disorders by the lipid peroxidation of cell membranes. Dysfunctional cell membranes may lead to increased permeability of the gut barrier and BBB, enabling microbial translocation from the GI tract into the systemic circulation, eventually reaching the brain.

Neurolipidomics is a rapidly growing field made possible by the recent advances in mass spectrometry (MS), a technique capable of quickly processing biomolecules, identifying in seconds the lipid species involved in various CNS pathologies [166,167]. Neurolipidomics has facilitated the development of receptor-independent antipsychotic treatments (RIATs), including LRT.

Lipid replacement therapy (LRT) is a technique that utilizes healthy, natural glycerophospholipids to substitute oxidized components of the plasma membranes lipid bilayer, restoring the physiological fluidity as well as neurotransmitter signaling. This approach, based on the oral supplementation with natural phospholipids and antioxidants, was demonstrated to halt the dissemination of cellular senescence to the neighboring, healthy cells, probably by inhibiting SASP [168,169,170,171,172]. Moreover, LRT was shown to supplant not only cell membranes but also the damaged mitochondrial inner and outer membranes with new and natural lipid species. Indeed, loss of lysophosphatidylethanolamine (LPE), phosphatidylglycerol (PG), and phosphatidylinositol (PI) in senescent mitochondria was shown to cause organelle demise; conversely, replacement with healthy lipids promotes mitochondrial thriving [171]. 

## 11. Phosphoinositide-Dependent Kinase 1 (PDK-1) Inhibitors

We surmise that combining LRT with inhibitors of phosphoinositide-dependent kinase 1 (PDK-1), such as kaempferol, would produce superior results in SCZ compared to LRT alone. Indeed, PDK1 inhibitors reverse cellular senescence, a phenotype previously considered irreversible. Lowered SASP together with the healthy exogenous lipids decrease the risk of peroxidation and ferroptosis [172]. Interestingly, kaempferol is also an antagonist of AhR, a transcription factor and cellular senescence driver. On the other hand, PDK1 activation of protein kinase B (Akt) and glycogen synthase kinase 3 beta (GSK-3β) promotes SCZ pathogenesis [173] (Figure 3). The point of contact between exogenously administered phospholipids and PDK1 is phosphatidylserine (PS), which binds PDK1 through its PH domain, blocking PS externalization and the initiation of cellular senescence. Moreover, PH–PS attachment inhibits PDK1 activation of downstream kinases, accounting for the antipsychotic actions of PDK1 inhibitors [174] (Figure 4, Table 3).

Taken together, LRT inhibits PDK1 as well as Akt and GSK-3β, exerting antipsychotic effects by entirely natural means.

To the best of our knowledge, we are the first to recommend LRT with PDK1 inhibitors based on the efficacy of each individual intervention. LRT and PDK1 inhibitors are natural compounds that are taken orally. Several studies found that LRT can avert the release of SASP by senescent cells, reducing their ability to disseminate this phenotype. Other studies discuss the potential use of PDK-1 inhibitors, such as kaempferol (a flavonoid), for SCZ and AD (Table 3).

## 12. Recombinant Human Interleukin-22 (IL-22)

Hinting at the microbiome, Emil Kraepelin stated that SCZ was caused by toxins generated in various body regions that could eventually reach the brain, triggering psychosis [66,183]. In addition, the membrane phospholipid hypothesis of SCZ, developed in the 1990s, foreshadowed LRT by predicting that cell membranes can be rehabilitated by exogenous phospholipids [184]. 

Microbial translocation from the GI tract into the host circulatory system was proposed over 100 years ago but completely forgotten after Kraepelin’s death. Along this line, the *Scientific American* from October 1896, volume 75, issue 15, ran the story “Is Insanity Due to a Microbe?”, highlighting bacteria-mediated neuropathology, the model supported by the scientists of that time (Figure 5).

During the HIV epidemic of the 1980s, microbial translocation, a common phenomenon encountered in this disease, was linked to virus-depleted IL-22, a cytokine considered “the guardian” of the gut barrier [185]. The same pathology was reported during the COVID-19 pandemic, emphasizing further that viruses and bacteria can trigger psychosis [186,187]. For this reason, we believe that supplementation with exogenous IL-22 can restore the integrity of gut barrier, ameliorating the symptoms of both SCZ and dementia. Recombinant human interleukin-22 (IL-22), currently being developed as a cancer therapy, is a cytokine known for protecting the biological barriers, iron downregulation, and stimulation of adult neurogenesis, suggesting a beneficial role in SCZ [187]. 

## 13. Aryl Hydrocarbon Receptor (AhR) Antagonists

AhR is a cytosolic transcription factor, initially associated with dioxin toxicity, which responds to numerous endogenous and exogenous stimuli, including xenobiotics [188,189]. In the cytoplasm, AhR is anchored by two molecules of heat shock protein 90 (HSP90) and upon detachment from this chaperone, it enters the nucleus to activate or silence the transcription of many genes. Aside from dioxin, AhR binds several ligands relevant for neuropsychiatry, including DA, serotonin, and the drug clozapine, bringing this receptor into the psychopharmacological arena [190,191] (Figure 6). In contrast, aripiprazole augments the HSP90 link, preventing AhR dissociation and nuclear entry, evidencing AhR antagonist effects [192].

Dietary AhR antagonists, such as resveratrol, luteolin, kaempferol, quercetin and the synthetic compounds CH223191, alpha-naphthoflavone, 6-bromo-3’-nitroflavone, BAY2416964, and HBU651 are beneficial for lowering cellular senescence, ameliorating psychosis, and cognition [193,194]. Furthermore, AhR exerts antagonistic pleiotropy (positive effects during the development, and detrimental ones later in life). Pleiotropy is responsible for the dioxin property of activating AhR in adulthood, inducing premature cellular senescence and cognitive impairment [195,196,197]. 

## 14. Conclusions

While acute psychosis responds very well to antipsychotic drugs, chronic psychotic illnesses are much more refractory to these agents. Over the past decades, psychopharmacology has focused excessively on the receptor-dependent actions of antipsychotic drugs and put much less emphasis on the receptor-independent ones, such as antimicrobial and anticancer actions. These are significant for the elimination of translocated microbes and de-escalation of immune responses directed at microbial molecules, such as LPS or mcfDNA. The neglect of noncanonical action mechanisms of antipsychotic drugs over the past decades has severely limited the development of new models and therapies, leaving chronic psychotic illnesses dependent on the treatments for acute psychosis. 

The long-term exposure to conventional antipsychotic drugs may have detrimental effects on cell membrane lipidome by triggering peroxidation and intercalation into the lipid bilayer. In turn, this may lead to neuronal damage and gray matter loss documented by neuroimaging studies. To avert neuronal loss by cell membrane damage, elderly people with SCZ should receive LRT and PDK1 inhibitors in conjunction with conventional antipsychotics.

Exogenous human recombinant IL-22 protects intestinal epithelia, preventing the microbes or their components from translocating outside the GI tract. This, in turn, averts aberrant immune system activation and the subsequent end-organ damage. Moreover, AhR inhibitors reverse the antagonistic pleiotropy by blocking late-life activation of this protein.

Receptor-independent antipsychotics are likely beneficial for the PLWS and comorbid dementia, as these conditions are characterized by damaged lipidomes and cell membranes as well as increased intestinal permeability. Lipid supplementation and PDK1 inhibitors are candidates for future studies due to their potential to delay the onset of dementia. More studies are needed to evaluate the beneficial effects of these strategies as well as that of iron chelators for restoring cognitive capabilities in forensic detainees and, ultimately, their capacity to stand trial. For example, what neurolipidomic interventions are best suited for LRT? Which PDK1 inhibitor would lead to optimal Akt inhibition? Would phosphatidylserine supplementation be more effective in diffusing optimally through the membranes as it is situated on the inner leaflet of the cell membrane? What about mitochondrial transfer or transplant, would it improve schizophrenia outcomes [198]?

## Figures and Tables

**Figure 1 ijms-24-15797-f001:**
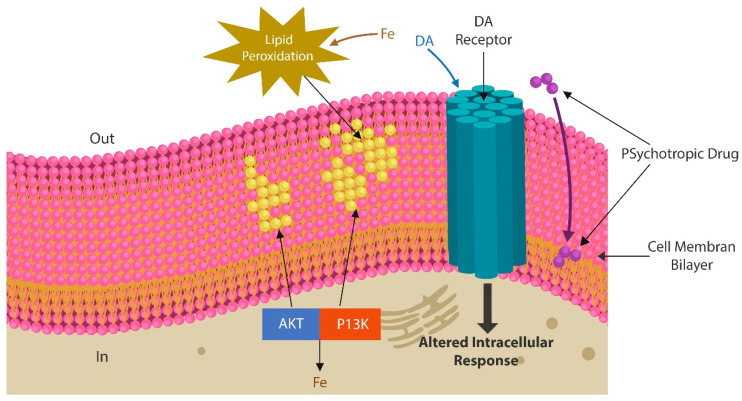
Neuronal membranes are comprised of a lipid bilayer that anchors numerous neurotransmitter receptors. Antipsychotic drugs and excessive iron trigger peroxidation of the lipid bilayer, altering membrane permeability (in yellow). Intracellular iron activates downstream SCZ-related kinases, including Akt and PI3K contributing to the pathogenesis of this disease. Psychotropic drugs, especially Phenothiazines, intercalate themselves into the lipid bilayer, changing the biophysical properties of plasma membranes, which, in turn, disrupts neurotransmission (purple).

**Figure 2 ijms-24-15797-f002:**
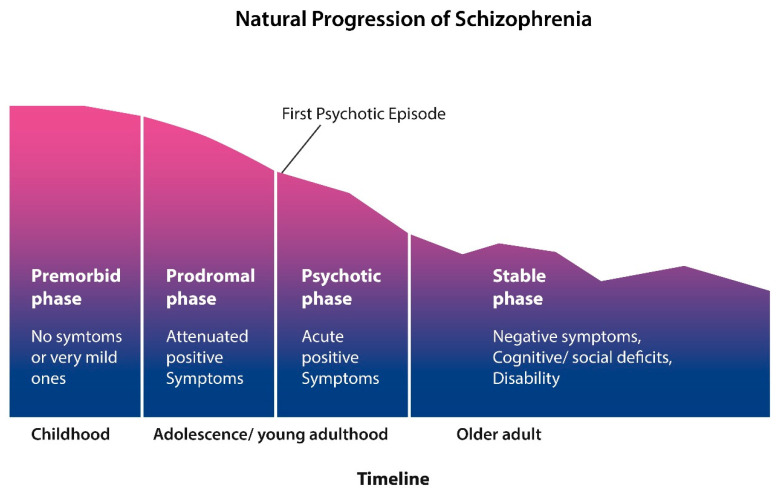
Natural course of SCZ; the disorder starts in early childhood with a premorbid phase with no symptoms or very mild ones. Prodromal phase is marked by attenuated symptoms, such as social isolation, anxiety, and insomnia. This phase blends gradually into the psychotic phase during which patients are usually hospitalized numerous times for exhibiting positive symptoms. Around midlife, the positive symptoms gradually subside and are replaced by the negative symptoms and cognitive manifestations. (Figure adapted from ref. [104]).

**Figure 3 ijms-24-15797-f003:**
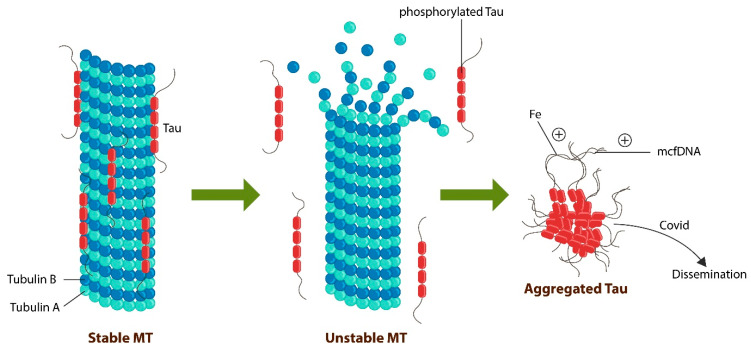
MTs are stabilized by the tau protein. Excessive tau phosphorylation detaches this protein, destabilizing the MTs (in red). Hyperphosphorylated tau aggregates, forming neurofibrillary tangles (NFTs), which accumulate, triggering neuronal death. Both iron and mcfDNA promote tau aggregation, while pathogens, including the SARS-CoV-2 virus, disseminate pathological tau to neighboring healthy cells, promoting the development of dementia.

**Figure 4 ijms-24-15797-f004:**
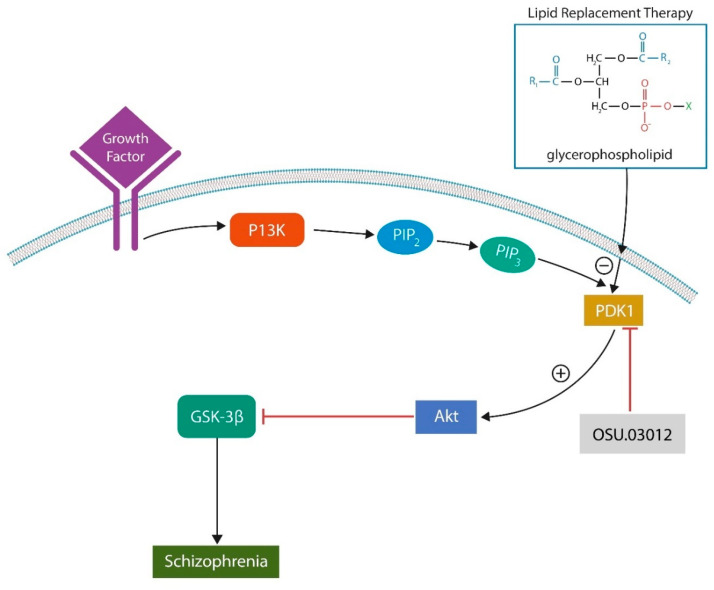
LRT inhibits PDK1 as exogenous phosphatidylserine (PS) binds the PH domain of PDK1, inactivating the downstream kinases, protein kinase B (Akt), and glycogen synthase kinase 3 beta (GSK-3β). LRT and PDK1 inhibitors act as natural antipsychotic agents. The novel PDK1 inhibitor OSU 03012 readily crosses the BBB, enhancing the antipsychotic properties of LRT.

**Figure 5 ijms-24-15797-f005:**
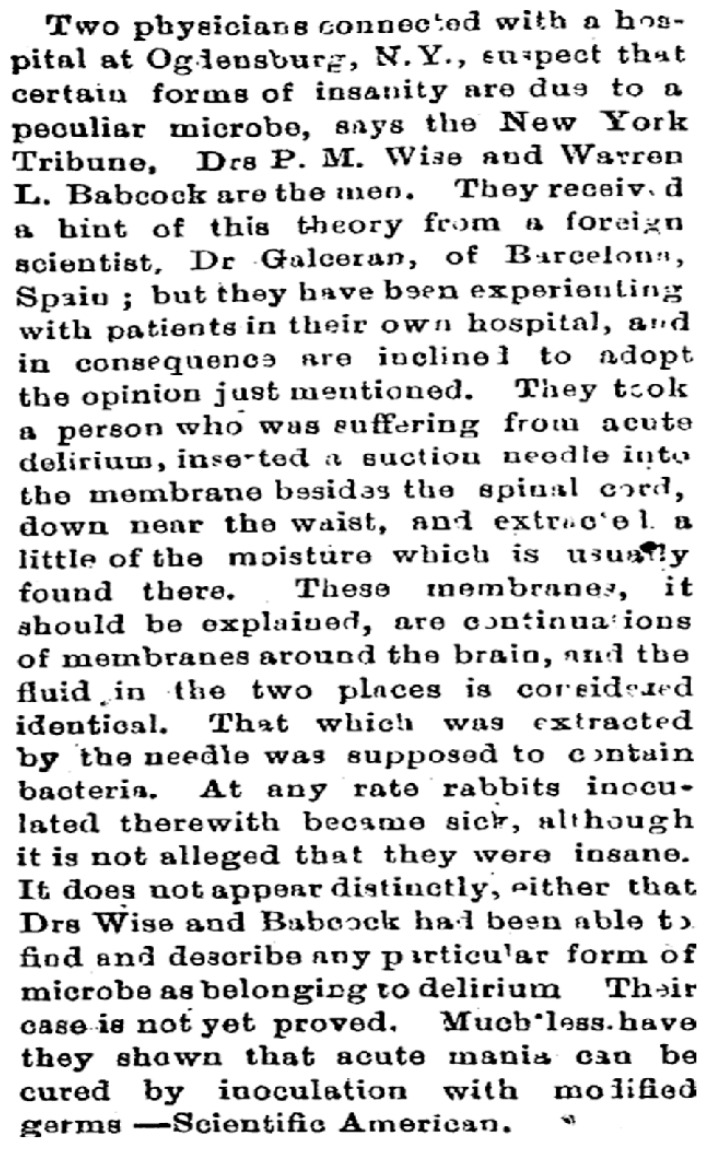
Endorsed by Emil Kraepelin and other 19th century researchers, microbial hypothesis of. SCZ was forgotten until the arrival of HIV and COVID-19.

**Figure 6 ijms-24-15797-f006:**
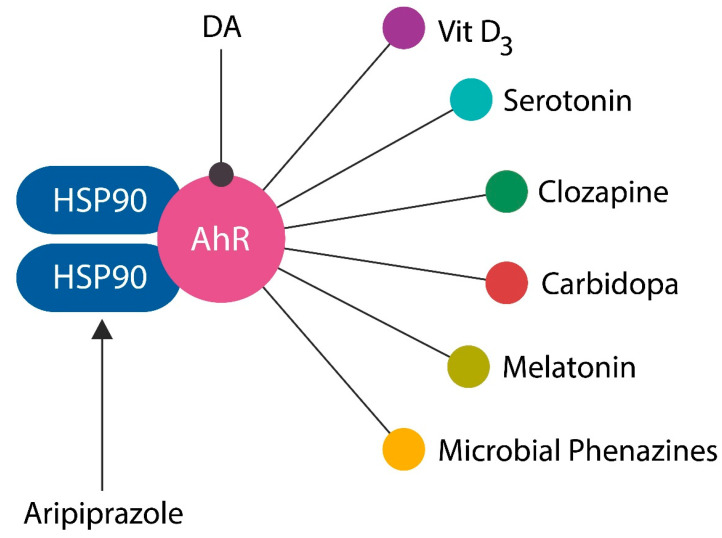
AhR, a receptor for several neurotransmitters, including DA, serotonin, melatonin, and vitamin D3, as well as clozapine, carbidopa, and microbial phenazines (natural phenothiazines). Aripiprazole enhances HSP90 attachment to AhR, blocking its entry into the nucleus and the activation of SCZ vulnerability genes.

**Table 1 ijms-24-15797-t001:** Sustained recovery of SCZ patients during the 20th century, including employment (table adapted from Warner R [54]).

Interval	Sustained Recovery	Employed
1901–1920	20%	4.7%
1921–1940	12%	11.9%
1941–1955	23%	4.1%
1956–1975	20%	5.1%
1976–1995	20%	6.9%

**Table 2 ijms-24-15797-t002:** Standardized tests for dysfunctional gut barrier and translocation.

Marker Type	Marker Assay	References
Integrity of gut barrier	IL-22 Singulex-Erenna^®®^	[127]
Translocated microbes	mcfDNA Karius Test^®®^	[139]

**Table 3 ijms-24-15797-t003:** Natural flavonoids PDK1 inhibitors.

PDK1 Inhibitor	Plant	References
Kaempferol	Fruits, vegetables, and herbs	[175]
Quercetin	Onions, kale, broccoli	[176]
Myricetin	Oranges, berries, tomatoes, nuts, tea	[177]
Epigallcatechin-3 gallate	Green tea	[178]
Lupiwighteone isoflavone	Glycyrrhiza glabra; Lotus pedunculatus	[179]
Delphinidin	Citrus fruits	[180]
Honokiol	Cherries, berries, grapes	[181]
Delphinidin	Cranberries, concord grapes, pomegranates	[182]

## Data Availability

Not applicable.

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
