# Peer review of "Receptor-Independent Therapies for Forensic Detainees with Schizophrenia–Dementia Comorbidity"

_ijms, 2023, doi:10.3390/ijms242115797_

Round 1
Reviewer 1 Report
Comments and Suggestions for Authors
This article aims to represent the unmet needs in forensic detainees with different severe psychiatric illnesses, such as schizophrenia and dementia, and provide possible new avenues for treatment other than antipsychotics. This is an important topic. However, there are major issues in the article.
Please, provide what is the type of this review article (i.e., a narrative review). Please, specify methods, and what were the key words for literature search. Articles usually start with the abstract, followed by the introduction. The present article has several sentences before the abstract, and then between the abstract and introduction.
Introduction
The first section has no references cited, please, add references to support claims
Antipsychotics are described as harmful, because of receptor-dependent and receptor-independent pharmacological action, which encourages the invention of new agents. The authors list studies to support unwanted effects of antipsychotics. However, did all studies confirm those findings unequivocally, such as gray matter loss, peroxidation of lipids in plasma membranes and antimicrobic and antiproliferative effects, etc...? Are those effects dose-dependent? Do low doses also have such untoward effects?
Please, provide data on the relationship between anosognosia and poor illness insight.
„Due to improved care over the past three decades, the lifespan of this population has increased considerably “-is this only due to improved care? There are several meta-analyses (Correll CU, Solmi M, Croatto G, Schneider LK, Rohani-Montez SC, Fairley L, Smith N, Bitter I, Gorwood P, Taipale H, Tiihonen J. Mortality in people with schizophrenia: a systematic review and meta-analysis of relative risk and aggravating or attenuating factors. World Psychiatry. 2022 Jun;21(2):248-271. doi: 10.1002/wps.20994; Jia N, Li Z, Li X, Jin M, Liu Y, Cui X, Hu G, Liu Y, He Y, Yu Q. Long-term effects of antipsychotics on mortality in patients with schizophrenia: a systematic review and meta-analysis. Braz J Psychiatry. 2022;44(6):664-673. doi: 10.47626/1516-4446-2021-2306; Taipale H, Tanskanen A, Mehtälä J, Vattulainen P, Correll CU, Tiihonen J. 20-year follow-up study of physical morbidity and mortality in relationship to antipsychotic treatment in a nationwide cohort of 62,250 patients with schizophrenia (FIN20). World Psychiatry. 2020;19(1):61-68. doi: 10.1002/wps.20699.), reporting that antipsychotics actually decrease mortality in schizophrenia, so please discuss how can this been explained in respect with adverse impact of antipsychotics on cellular senescence.
Table 1, adapted from the data of the review Warner et al, 2009, suggests that „the recovery rate following the introduction of antipsychotic drugs was not very different compared to the pre-psychopharmacological era “, and authors conclude that: „Taken together, SCZ outcome studies suggest that the long-term evolution of chronic psychotic disorders leads to disability and cognitive deficits, regardless of treatment with DA-blocking agents “. To support such conclusion, more data are needed. While recovery may still be an ambitious goal to achieve in schizophrenia, there are other more achievable outcomes such as remission, or at least partial remission. So, please, provide data whether rates of remission have changed since Kraepelin time, and also the lengths of hospitalizations, and the proportion of patients who were institutionalized. Otherwise, this text leads to conclusion that antipsychotics are not effective in schizophrenia, and that patients are as psychotic and disabled as they were 100 years ago, which is simply not the case. Antipsychotics are far from being perfect, but are still the cornerstone of treatment. The authors also recognize this, given the statement that „The currently available antipsychotic drugs are extremely effective for the treatment of acute psychotic disorders... “on page 10.
In addition, „For example, a century ago, there were large public institutions for tuberculosis, leprosy, and mental illness, while today only the latter remain in existence “. This is because tuberculosis and leprosy are infectious diseases and could be eradicated by antimicrobial agents. On the contrary, the origin of schizophrenia is unknown, and thus its cause cannot be treated. The treatment of schizophrenia is symptomatic, and cannot be compared with disorders caused by microorganisms.
The authors conclude that „To avert neuronal loss by cell membrane damage, elderly with SCZ should receive LRT and PDK1 inhibitors in conjunction with conventional antipsychotics, and „Elderly forensic detainees in treatment with conventional antipsychotic drugs should receive lipid supplementation and PDK1 inhibitors to delay the onset of dementia “. Please, provide evidence. Are there clinical studies on the efficacy of LRT and PDK1 inhibitors in patients with schizophrenia? Based on what can such treatment be recommended? Please, provide results preclinical and clinical data
Reviewer 2 Report
Comments and Suggestions for Authors
1. It would be nice if the writers could provide an explanation for several scientific terms, such as "interoceptive awareness" and "anosognosia," when they first arise. Also, I’d like to the authors emphasize their roles in different pathological conditions and their implications for behavior.
2. The authors discuss "molecular underpinnings of cellular cognition" but don't go into detail. It would be beneficial to extend the conversation by providing insights or examples of current research on molecular pathways relating brain dysfunction to behavior.
3. Could the authors provide more supporting evidence for the hypothesis “anosognosia begets psychosis” and discuss its implications for understanding psychiatric disorders?
4. It is not widely understood, in my opinion, that schizophrenia is linked to cellular aging. So I'd want to read a bit more conceptual foundation of schizophrenia as a sort of segmental progeria rooted in the existing body of literature. A more detailed description of the theoretical underpinnings that lead to this conception could also be beneficial.
5. The authors stated that the reduced average longevity of schizophrenia patients is a result of the accelerated cellular aging. I'm not certain if this is correct. For example, I believe schizophrenia patients are at a higher risk of cardiovascular disease, which may contribute to their shorter lifespan. Also, if there are publications that support the authors' claim, it would be useful if the writers could offer that evidence.
6. In the last sentence on page 4, it says, "We believe that, because of their great size, VENs are... - 1) Could you please clarify whether the claimed causal relationships are supported by research already in existence or whether they are just speculative? For example, are there studies demonstrating that VENs are more susceptible to lipid peroxidation, or is this a novel idea? Exists proof of Tau hyperphosphorylation in bvFTD and SCZ VENs? 2) Regarding the potential treatments, is there any existing evidence of their efficacy in bvFTD or related conditions?
7. “Taken together, SCZ outcome studies suggest that the long-term evolution of chronic
psychotic disorders lead to disability and cognitive deficits” The authors should consider that most of the current available anti-psychotic treatment are mainly targeting the positive symptoms while the treatment for cognitive deficits in schizophrenia is underdeveloped.
8. The draft is quite long and there were instances where I found it challenging to follow the flow of information. For enhanced readability, the authors might consider organizing the content with section titles and subtitles, if the journal's guidelines permit.
9. "Elderly with SCZ should receive LRT and PDK1 inhibitors in conjunction with conventional antipsychotics” even this is in conclusion paragraph, it is important to clarify whether this is based on established guidelines, clinical trial data, or is a novel suggestion based on your review.
10. “More studies are needed to evaluate the beneficial….” Could you be more specific about the type of studies needed based on your review?
Minor comments:
1. “The latter category includes patients with frontotemporal dementia behavioral variant (bvFTD), a condition with clinical manifestations similar to those of schizophrenia (SCZ).” It’d be great if the authors could briefly introduce the overlapping symptoms which may help the reader to understand the background.
2. “As the incidence of bvFTD has increased dramatically in forensic population over the past two decades,” I believe it’d be helpful to provide the exact estimates with citations. This comment also applies to all similar statements throughout the main text.
3. Abbreviations: 1) “late-life dementia in people living with SCZ (PLWS)” I’d suggest the authors to avoid unnecessary and uncommon abbreviations. 2) “mcfDNA” please provide the full name upon its first appearance.
4. Please correct typos such as “corelated” in “ …be directly corelated to a specific central nervous system (CNS) region or pathology,…”
Comments on the Quality of English Language
In my opinion, this draft used a bit too many abbreviations, including some of them are uncommon and unnecessary. Also, please remember to define and clarify all technical terms at their first occurrence.
Round 2
Reviewer 1 Report
Comments and Suggestions for Authors
Please, provide methods, i.e., keywords and databases used in search for references. Please describe and add IN TEXT all references which led to the conclusion: "Elderly forensic detainees in treatment with conventional antipsychotic drugs should receive lipid supplementation and PDK1 inhibitors to delay the onset of dementia". Add in text all clinical trials which have establish the efficacy. The recommendations are based on clinical trial results. Otherwise, only future trials could be recommended
Round 3
Reviewer 1 Report
Comments and Suggestions for Authors
Authors have done changes in the text. However, I am still concerned with the sentence in the conclusion: Elderly forensic detainees in treatment
with conventional antipsychotic drugs should receive lipid supplementation and PDK1 inhibitors to delay the onset of dementia. Those agents are not approved for the prevention of dementia.
I do not see Table 2 Natural, oral LRT and PDK1 inhibitors we recommend for elderly with SCZ in treatment with antipsychotic medications in the text. There is, however, a list of references. Instead simply providing the references, please, provide short description about the most important findings of those references, and provide reference number in the parenthesis, as usually done in the IJMS articles
Please, delete the word "antipsychotic" from the title: Receptor-independent antipsychotic therapies for forensic detainees with schizophrenia-dementia comorbidity, because this is not antipsychotic therapy. The only antipsychotic therapy so far are only D2-blocking agents and 5-HT2A blocking drug pimavanserin, nothing else is considered antipsychotics.
Figure 4. LRT and PDK1 inhibitors act as natural antipsychotic agent - please, provide references
Please, provide information in which form are LRT and PDK1 inhibitors available on the market, and what are adverse events of this treatment
Round 4
Reviewer 1 Report
Comments and Suggestions for Authors
This is an interesting and important topic, and authors made some changes in the text as suggested, but:
1) I still do not agree with the statement that: Elderly forensic detainees in treatment with conventional antipsychotic drugs should receive lipid supplementation and PDK1 inhibitors to delay the onset of dementia. There is no sufficient evidence (randomized studies) to claim this. It needs to be replaced with something like: lipid supplementation and PDK1 inhibitors are candidates for future studies due to their potential...
2) The text in "answers to reviewers" and references provided should be also incorporated in the text. It explains the rationale for considering lipid supplementation and PDK1 inhibitors in clinical practice and for future studies
